# Co-Inheritance of Pathogenic Variants in *PKD1* and *PKD2* Genes Determined by Parental Segregation and De Novo Origin: A Case Report

**DOI:** 10.3390/genes14081589

**Published:** 2023-08-06

**Authors:** Ludovico Graziani, Stefania Zampatti, Miriam Lucia Carriero, Chiara Minotti, Cristina Peconi, Mario Bengala, Emiliano Giardina, Giuseppe Novelli

**Affiliations:** 1Department of Biomedicine and Prevention, University of Rome “Tor Vergata”, 00133 Rome, Italy; miriamcarriero@gmail.com (M.L.C.); chiara.minotti95@gmail.com (C.M.); emiliano.giardina@uniroma2.it (E.G.); novelli@med.uniroma2.it (G.N.); 2Genomic Medicine Laboratory UILDM, IRCCS Fondazione Santa Lucia, 00179 Rome, Italy; s.zampatti@hsantalucia.it (S.Z.); cpeconi@alice.it (C.P.); 3Medical Genetics Unit, Tor Vergata University Hospital, 00133 Rome, Italy; mario.bengala@ptvonline.it

**Keywords:** ADPKD, *PKD1/2*, digenic disease, genotype–phenotype correlations, NGS

## Abstract

Autosomal dominant polycystic kidney disease (ADPKD) is the most common hereditary renal disease, and it is typically caused by *PKD1* and *PKD2* heterozygous variants. Nonetheless, the extensive phenotypic variability observed among affected individuals, even within the same family, suggests a more complex pattern of inheritance. We describe an ADPKD family in which the proband presented with an earlier and more severe renal phenotype (clinical diagnosis at the age of 14 and end-stage renal disease aged 24), compared to the other affected family members. Next-generation sequencing (NGS)-based analysis of polycystic kidney disease (PKD)-associated genes in the proband revealed the presence of a pathogenic *PKD2* variant and a likely pathogenic variant in *PKD1*, according to the American College of Medical Genetics and Genomics (ACMG) criteria. The *PKD2* nonsense p.Arg872Ter variant was segregated from the proband’s father, with a mild phenotype. A similar mild disease presentation was found in the proband’s aunts and uncle (the father’s siblings). The frameshift p.Asp3832ProfsTer128 novel variant within *PKD1* carried by the proband in addition to the pathogenic *PKD2* variant was not found in either parent. This report highlights that the co-inheritance of two or more PKD genes or alleles may explain the extensive phenotypic variability among affected family members, thus emphasizing the importance of NGS-based techniques in the definition of the prognostic course.

## 1. Introduction

Autosomal dominant polycystic kidney disease (ADPKD; MIM# 173900 and 613095) is the most common hereditary kidney disease and one of the most frequently inherited disorders in humans, with an estimated prevalence of 1/400 to 1/1000 in the general population [1,2]. It is predominantly characterized by the development and progressive enlargement of cystic formations in both kidneys. Cysts can arise in other organs, including the liver, pancreas, and spleen, with Polycystic liver disease (PLD) being the most common extra-kidney manifestation of ADPKD [3]. Other extrarenal manifestations include intracranial and aortic aneurysms and cardiovascular abnormalities [4,5].

Renal manifestations of ADPKD are directly related to the process of cystogenesis and include progressive decline in kidney function, hypertension, and kidney pain. Affected individuals usually present in the third and fourth decade and develop end-stage renal disease (ESRD) in late middle age [4,6]. The clinical manifestations can also include hematuria, urinary tract infection, and urolithiasis, as renal cysts can be complicated by hemorrhage, rupture, infection, and nephrolithiasis [3].

Renal cysts originate from the epithelia of nephrons and renal collecting system and show an abnormally high rate of cellular proliferation [7]. Alterations in cell growth and differentiation, fluid secretion, and extracellular matrix composition have been related to cystogenesis, yet polycystic kidney disease (PKD) mechanisms remain incompletely understood [1,8,9,10,11,12].

ADPKD is genetically heterogeneous and is mainly caused by germline variants in either polycystic kidney disease 1 (*PKD1*) (MIM* 601313; located on chromosome 16p13.3) or polycystic kidney disease 2 (*PKD2*) (MIM* 173910; located on chromosome 4q21) genes, accounting for ~78% and ~15% of cases, respectively [3]. Heterozygous pathogenic variants in additional genes, such as *GANAB* and *DNAJB11*, were recently shown to cause polycystic kidneys only in a small proportion of affected subjects [13,14]. *PKD1* encodes Polycystin-1 (PC-1), a large glycoprotein anchored to the cellular membrane by 11 transmembrane helices and expressed in a variety of tissues, including kidney, brain, heart, bone, and muscle [15,16]. Although the function of Polycystin-1 is not well understood, it is speculated that the protein may act as a cell surface receptor for an as yet unidentified extracellular ligand [5]. The *PKD2* gene product is Polycystin-2 (PC-2), an integral membrane protein that is widely expressed in different tissues, such as kidney, heart, ovary, testis, vascular smooth muscle, and small intestine [17,18]. PC-2 shares structural features with Polycystin-1 and the transient receptor potential polycystic (TRPP) family of calcium-regulated cation channels. Several studies have shown that PC-2 functions as a divalent cations channel and that it can increase cytosolic calcium [5,19,20,21].

Several subcellular localizations have been suggested for polycystins, including primary cilia, which are thought to be key to polycystic kidney disease (PKD) pathogenesis [22,23]. Although it is not clear how alterations of ciliary function would lead to cyst formation in PKD, previous studies have shown that the bending of renal cilia stimulates a rise in intracellular calcium concentration, suggesting that cilia may function as mechanosensors of urine flow [24]. 

Recent studies indicate that PC-2 directly interacts with PC-1 and that the polycystin complex is associated with several cell signaling pathways, including the regulation of calcium homeostasis mediated by the cAMP pathway [1,25,26]. PC-1 and PC-2, together with fibrocystin/polyductin, which is responsible for autosomal recessive polycystic kidney disease (ARPKD) (MIM# 263200), form a heteromeric complex thought to function as a receptor channel [23,27]. The interaction between both polycystins in a common pathway would explain why pathogenic variants in *PKD1* and *PKD2* produce similar clinical manifestations. 

Most individuals diagnosed with ADPKD have an affected parent, whereas in 10% to 20% of cases the disorder is the result of a de novo pathogenic variant [3,28]. ADPKD is transmitted in an autosomal dominant, fully penetrant pattern of inheritance, i.e., virtually all individuals carrying a PKD variant in their germline will develop multiple bilateral cysts within the kidneys in mid to late adulthood [29]. However, the clinical spectrum of both renal and extrarenal manifestations of the disease, in terms of age at diagnosis, renal survival and complications, is often significantly variable among affected individuals, even within a single family [3,4,30]. Intra- and inter-familial disease severity reportedly ranges from onset in prenatal/neonatal period in very early onset ADPKD (ADPKD_VEO_) and before the age of 15 years in early onset ADPKD (ADPKD_EO_) to elderly patients with preserved kidney function [31]. Pathogenic variants in *PKD1* are typically associated with earlier onset and more severe disease than in *PKD2-*patients, with an average age at ESRD of 54 in *PKD1*-related ADPKD, versus 74 years in *PKD2*-related disorders [2,5]. Moreover, protein-truncating variants in *PKD1* and *PKD2* genes are associated with more severe disease, leading to ESRD earlier in life than non-truncating variants [1,3,32]. 

The genetic heterogeneity caused by the *PKD1* and *PKD2* genotypes cannot, however, solely explain the extreme phenotypic variability observed, suggesting that a more complex pattern of inheritance is possible [33]. Unusual complex genotypes, due to additional genetic defects, such as variants in multiple PKD-related genes or biallelic *PKD1* or *PKD2* variants in trans and/or de novo cases, have been reported in a minor subset of patients with very early onset and much more severe disease, suggesting they are likely to aggravate the phenotype [1,29,34,35,36].

Herein, we report the first case of digenic inherited/de novo ADPKD in an Italian family, in which the more severely affected member carries a *PKD1* causative allele in addition to the familial *PKD2* germline variant.

## 2. Case Presentation

The proband (III-2), a female, is the second child of Italian non-consanguineous parents (Figure 1). She was addressed to our institution at the age of 34 years by the attending physician due to a clinical suspicion of ADPKD. At 14 years of age, she underwent abdominal ultrasound (US) which showed enlarged kidneys with multiple cysts and multiple hepatic cysts. Further investigations, such as a basal electrocardiogram and echocardiography, were normal. Physical and phenotypic examinations showed no external anomalies. The patient attended school normally and successfully completed her studies. 

ADPKD was diagnosed based on US and familiarity (ADPKD had been found in her father, II-1). Chronic kidney disease progressed to End-Stage Renal Disease (ESRD) at 24 years of age. Before kidney transplantation, hematochemical values of routine blood exams showed electrolytes within a normal range (Na+ 140 mmol/L, K+ 5.0 mmol/L), increased uric acid (9.6 mg/dL), hyperphosphatemia (8.0 mg/dL), hyperazotemia (236 mg/dL), hypercreatininemia (6.37 mg/dL), and eGFR (Estimated Glomerular Filtration Rate) was 8 mL/min/m^2^, according to the CKD EPI (Chronic Kidney Disease Epidemiology Collaboration) formula [37] (Table 1). She underwent living donor transplantation at the age of 24. 

Not much information is available on the paternal medical history. Abdominal US at 52 years of age showed enlarged kidneys with multiple cysts and a simple hepatic cyst. The father underwent kidney transplantation at the age of 62. Before kidney transplantation, hematochemical tests indicated hyponatremia (Na+ 132 mmol/L), hyperkalemia (K+ 6.18 mmol/L), hyperphosphatemia (6.7 mg/dL), hyperazotemia (234 mg/dL), hypercreatininemia (13.49 mg/dL), and eGFR was 3.5 mL/min/m^2^, according to the CKD EPI formula [37] (Table 1). 

A clinical diagnosis of ADPKD had also been made in three paternal aunts (II-5, II-6, II-7), in one paternal uncle (II-3), and in the paternal grandfather (I-1), who underwent kidney transplantation at 66 years of age, consistent with an autosomal dominant transmission pattern. To the best of our knowledge, none of the affected family members had evidence of intracranial arterial aneurysms.

After genetic counselling, informed consent was obtained from the proposita and her parents, who underwent diagnostic molecular tests. Genomic DNA of the trio was extracted from peripheral blood samples using the MagPurix Blood DNA Extraction Kit and MagPurix Automatic Extraction System (Resnova) (Catalog number ZP02001-48). We performed a whole exome next-generation sequencing (NGS) analysis (Illumina NextSeq 550 sequencer) and evaluated a gene panel for polycystic kidney disease (*PKD1*, *PKD2*, *GANAB*, *DZIPIL*, *PKHD1* and *DNAJB11*); potentially involved variants were confirmed through PCR amplification and Sanger sequencing. The NGS analysis revealed 675 SNVs and indel variants, of which 580 were low-coverage variants, 93 were benign or potentially benign, and 2 were potentially involved variants according to the American College of Medical Genetics and Genomics (ACMG) guidelines (Appendix A). In particular, the analysis showed a nonsense variant in *PKD2* (NM_000297.3: c.2614C>T; p.Arg872Ter) (rs755226061), inherited from the affected father. This variant was classified as “pathogenic” according to the ACMG guidelines [38]. The NGS analysis also demonstrated a de novo frameshift variant in *PKD1* (NM_001009944.2: c.11494_11495del; p.Asp3832), which was classified as “likely pathogenic” according to the ACMG guidelines [38].

## 3. Discussion

We report the first digenic ADPKD case determined by the combination of a de novo variant in the *PKD1* gene and an inherited variant in the *PKD2* gene, in a patient with early onset disease (diagnosed before 15 years old).

Heterozygous pathogenic variants in *PDK1* and *PKD2* genes are classically associated with ADPKD, accounting for ~78% and ~15% of cases, respectively [3]. 

Wide inter- and intra-familiar phenotypic variability is well-described in the scientific literature. Nonetheless, recent evidence has shown a rising complexity in the inheritance patterns for ADPKD; precocious and more severe clinical courses are often attributed to the concurrent effect of additional PKD alleles (inherited and/or de novo) in trans with a disease-causing variant [29,34]. 

The homozygosity of damaging variants either in *PKD1* or in *PKD2* genes is embryologically lethal, as demonstrated in knock-out murine models [39]. Mid-pregnancy spontaneous fetal loss has been described in association with compound heterozygosity of fully penetrant pathogenic variants in the *PKD1* gene, bilineally inherited from affected parents [40,41]. Nonetheless, biallelic inheritance can be, in some circumstances, compatible with life, whether either or both variants act as hypomorphic, as is the case with incompletely penetrant non-truncating variants determining reduced levels of functional protein. Therefore, additional variants may not themselves have a causative effect, but rather exert a disease-modifying role.

The combination of a hypomorphic variant in trans with a fully penetrant pathogenic variant has been previously documented in several ADPKD_VEO_ cases (disease diagnosed in utero or before 18 months). Biallelic incompletely penetrant *PKD1* alleles were described in patients with typical ADPKD-like phenotype 33 and in patients with early-onset disease and absent familiarity [29,32,35,42,43,44,45,46,47]. A single case of a neonatal-onset ADPKD, due to hypomorphic *PKD2* homozygosity, and that arose by uniparental disomy, has been reported to date [48]. In previous family studies, the coinheritance of a fully penetrant variant in either *PKD1* or *PKD2* genes or a different PKD-causing gene (e.g., *HNF1β* or *PKHD1*) was associated with earlier and/or more severe renal phenotypes [29].

To the best of our knowledge, the digenic genotype, including a fully penetrant variant in both *PKD1* and *PKD2* genes, has been rarely documented in the literature. 

Pei et al. (2001) described two siblings, both with a frameshift variant in the *PKD2* gene (p.Leu736Ter), and a second pathogenic missense variant within the *PKD1* (p.Tyr528Cys). Both patients presented with a more critical renal disease compared to the other family members who carried a single PKD variant, as they showed severely reduced creatinine clearance and manifested ESRD earlier (at the age of 48 and 52 years old, respectively). None of the affected family members had evidence of intracranial arterial aneurysms or massive polycystic liver disease [36,49]. 

A more recent case of digenic ADPKD_VEO_ and absent extrarenal-manifestations in a Czech family has been described by Elikasova et al. (2018). Two different PKD causative variants, i.e., *PKD1*:p.Gln2196Ter and *PKD2*:p.Arg420Gly, were inherited from affected parents, with milder clinical presentation. Chronic kidney disease progressed to ESRD at the age of 23 years old [1].

In the present report, no other family member has been genetically characterized in addition to the trio. Within the pedigree, no ADPKD_EO_ case was diagnosed except the main proband, and only two more individuals (the father and the grandfather of the proband) presented ESRD and underwent renal transplant, yet both in their sixties. However, the patient discussed had no evidence of intracranial arterial aneurysms or massive polycystic liver, resulting in either portal hypertension or hepatic failure. The loss-of-function variant identified both in the proband and her father, i.e., *PKD2* c.2614C>T p.(Arg872Ter), has been previously documented in the ClinVar (Accession SCV002767978.1; SCV002581298.1; SCV001523097.1; SCV001251249.1) and Human Gene Mutation Database (HGMD) (Accession CM994295) databases, as well as in the recent scientific literature, in association with ADPKD [50,51]. Nonetheless, the swift disease progression in our patient (radiological diagnosis before 15 years old and ESRD onset at 24 years old) was hardly explained by the presence of a single familial pathogenic variant in a PKD gene. Therefore, we first describe the novel frameshift variant c.11494_11495del (p.Asp3832ProfsTer128) in the *PKD1* gene, detected by NGS analysis. The variant occurring de novo in the proband is predicted to cause nonsense mediated decay (NMD) and was classified as “likely pathogenic” according to ACMG guidelines [38]. This additional finding is in accordance with the speculation that the most severe clinical course might be associated with a more complex genotype, thus underlining the importance of genotyping in the presence of pedigree with relevant clinical variability [31].

The diagnosis of ADPKD can be confirmed using well-established imaging criteria, i.e., through abdominal US, Computed Tomography (CT), and/or Magnetic Resonance Imaging (MRI), even in presymptomatic patients [2]. The molecular analysis of PKD-related genes currently plays a critical role in those patients whose diagnosis is uncertain (e.g., equivocal imaging results or sporadic cases) and when early disease exclusion is required (e.g., for reproductive counseling or kidney donation evaluation) [52]. However, as genetic testing increases in availability, also due to the advances of NGS-based methods, its application is expanding to the investigation of atypical clinical presentations (e.g., early/severe, or syndromic phenotype), to explain the genetic background of intrafamilial variability [53]. The discordant early onset ADPKD phenotype here described is explained by the occurrence of a de novo *PKD1* variant in addition to the paternal *PKD2* variant. We speculate that this complex genotype leads to a more severe phenotype, likely for a mechanism related to the interplay between PC-1 and PC-2 in the cystogenic process. Indeed, previous digenic knockout mouse models showed that PC-2 exerts a fundamental chaperone-like activity for the maturation and localization of the *PDK1* gene product [22], and PC-2 reduction was shown to determine a dose-dependent non-equimolar reduction in PC-1, which paralleled and further modified the renal phenotype [54]. A limitation of this study is the absence of adequate functional analysis involving the oligogenic *PKD1/PKD2* combination and disease. Further research and the development of tools and databases focusing on digenic inheritance are needed to advance our understanding of this genetic mechanism [33].

## 4. Conclusions

This is the first report of a digenic ADPKD patient, whose more complex renal phenotype is determined by the de novo occurrence of a fully penetrant variant in the *PKD1* gene, in addition to the hereditary burden in the *PKD2* gene. The co-inheritance of two or more ADPKD-related genes or alleles may explain the vast phenotypic variability, even among carriers of the same variant, through epistatic mechanisms and oligogenic effects. As the advances in genetic testing based on NGS technologies are expected to provide a clearer prognostic course, we suggest suspecting a complex underpinning genotype to investigate those ADPKD families where a more severe renal disease is hardly explained by a single segregating variant (i.e., in utero or early childhood). Nonetheless, a complex ADPKD genotype may not necessarily imply a more severe extrarenal involvement, even if further studies will be needed to better define the meaning of these observations.

## Figures and Tables

**Figure 1 genes-14-01589-f001:**
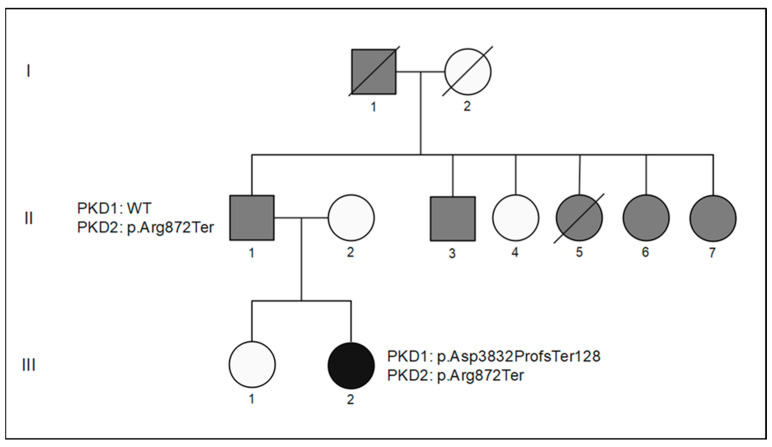
Segregation of the disease in the three-generation pedigree of the family, including the proposita (III-2) and her father (II-1). Gray-filled individuals are affected by Autosomal dominant polycystic kidney disease (ADPKD), presumably determined by the p.Arg872Ter variant in *PKD2* gene.

**Table 1 genes-14-01589-t001:** Clinical features and hematochemical tests obtained before kidney transplantation. Reference ranges are reported in brackets.

	Gender	Age at Admission	Natremia	Kalemia	Uricemia	Phospatemia	Azotemia	Creatininemia	eGFR ^1^	Age at Clinical Diagnosis	Age at Transplantation	Extra-Renal Manifestations
**Proband**	Female	34 years	140 mmol/L (136–145)	5 mmol/L (3.5–5.1)	9.6 mg/dL (2.4–5.7)	8.0 mg/dL (2.5–4.5)	236 mg/dL (16.6–48.5)	6.37 mg/dL (0.51–0.95)	8.0 mL/min/ 1.73 m^2^	14 years	24 years	Multiple hepatic cysts
**Father**	Male	63 years	132 mmol/L (136–145)	6.18 mmol/L (3.5–5.1)	NA	6.7 mg/dL (2.5–4.5)	234 mg/dL (16.6–48.5)	13.49 mg/dL (0.51–0.95)	3.5 mL/min/ 1.73 m^2^	52 years	62 years	Single simple hepatic cyst

^1^: according to CKD EPI (Chronic Kidney Disease Epidemiology Collaboration) formula [37]. NA: not available.

## Data Availability

All datasets used and/or analyzed during the current study are available from the corresponding author upon reasonable request.

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
