# Peer review of "Co-Inheritance of Pathogenic Variants in PKD1 and PKD2 Genes Determined by Parental Segregation and De Novo Origin: A Case Report"

_genes, 2023, doi:10.3390/genes14081589_

Round 1

Reviewer 1 Report

The paper submitted to ‘Gene’ is ‘Co-inheritance of pathogenic variants in PKD1 and PKD2 2

genes determined by parental segregation and de novo origin’ described a case report having a ADPKD.

1.     ADPKD is transmitted in an autosomal dominant, fully penetrant pattern of inheritance, however, the clinical spectrum of both renal and extrarenal manifestations of the disease is often significantly variable among affected individuals, even within a single family. The sentence seems incomplete. Can authors clarify why there are different manifestations when the disease is complete penetrance?

2.     The authors need to correct errors. Intra and inter-familial disease severity reportedly range from onset in prenatal/neonatal period (very early onset ADPKD, ADPKDVEO) and before age of 15 years (early onset ADPKD, ADPKDEO) to elderly patients with preserved kidney function1. Need to delete 1 here.

3.     The authors need to rewrite the introduction by shuffling some sentences to have a clear and fluent background reading about the ADPKD. The genetic heterogeneity caused by the PKD1 and PKD2 genotypes cannot solely explain the extreme phenotypic variability observed, suggesting that a more complex pattern of inheritance is possible. Additional genetic defects, such as variants in multiple PKD genes or biallelic PKD variants in trans and/or de novo have been reported in a minor subset of patients with very early onset and much more severe disease, suggesting they are likely to aggravate the phenotype. These sentences should come after the sentence mentioned in 1st comment in here.

4.     The case presentation can be in table format. The age, gender, height, clinical presentations such as creatinine, BUN, proteinuria index, and when the case admitted and follow-up time the case severity and features for easy understanding.

5.     Not much information is available on the paternal medical history. Can authors comment on the parents ADPKD status?

6.     Genomic DNA of proposita and their parents was extracted from peripheral blood sample using the MagPurix Blood DNA Extraction Kit and MagPurix Automatic Extraction System (Resnova). Need catalog numbers.

7.     We performed a whole exome NGS analysis (Illumina NextSeq 550 sequencer) and evaluated a gene panel for polycystic kidney disease. Need to present results in table format as the mutants versus WT gene panel sequences and amino acid changes. Need to state whether the data is readily available upon request or data deposited location.

8.     A more recent case of digenic ‘ADPKDVEO’ and absent extrarenal-manifestations. Was the ADPKDVEO is different than ADPKDEO?

Reviewer 2 Report

Review of the manuscript genes-2531502

Co-inheritance of pathogenic variants in PKD1 and PKD2 2 genes determined by parental segregation and de novo origin

The reviewed paper is a clinical case of a patient with a familial condition of autosomal dominant polycystic kidney disease (ADPKD), along with a discussion. The authors describe the description of a patient, presented with earlier and more severe renal phenotype (clinical diagnosis of the ADPKD at the age of 14 and end-stage renal disease aged 24), who was diagnosed with mutations in two main genes associated with the pathogenesis of APKD (PKD1 and KD2). The Authors revealed the presence of a pathogenic PKD2 variant and a likely pathogenic variant in PKD1. It is worth emphasizing that the mutation in the PKD1 gene was inherited, while in the PKD2 gene – was de novo. The authors concluded that this kind of a two-gene variant of genetic disorders predisposed to early onset of the disease.

I find the job interesting. However, I have a few comments before its eventual final publication:

Substantive remarks:

1. The current title does not precisely reflect the type of the paper. The manuscript is a case report and this should be clearly stated. The title should indicate that the paper is a case report of a patient with autosomal dominant polycystic kidney disease.

2. In the Introduction, however, the authors should describe the pathophysiology and symptomatology of ADPKD in more detail, together with the role of proteins - polycystins in the pathogenesis of the disease, which would be the starting point for the description of genetic issues.

3. Were the genetic tests of the entire patient's family described in the manuscript preceded by the approval of the appropriate institutional Bioethical Committee? It should be mentioned in the beginning of the section “Case presentation”.

4. How was GFR determined? (Table 2). What formula was the basis for estimating the eGFR value? Why is there no eGFR for the second patient - the father?

5. What methods were used to determine the values of the described biochemical parameters?

6. Discussion - it would be worth describing how many other cases of ADPKD have been described in the literature. Moreover, it should be discussed whether in other so far described cases, PKD1 or PKD2 disturbances have been more frequently reported?

7. Moreover, are there any other cases of ADPKD associated with other digenic PKD1/PKD2 mutations described by other authors in the literature?

TECHNICAL NOTES

8. The style of citation in the text is inconsistent with the guidelines of the Genes (MDPI Journal) - citations should not be in the form of footnotes, but in the form of citations in square brackets.

9. The style of the list of references is inconsistent with the guidelines of the Genes (MDPI Journal) - redundant numbers of the issues of the cited articles are given, no proper bolding of the year of publication, no italics in the journal abbreviations and volume numbers.

Round 2

Reviewer 2 Report

Re-review of the manuscript genes-2531502

Co-inheritance of pathogenic variants in PKD1 and PKD2 genes determined by parental segregation and de novo origin: a case report.

I would like to thank the Authors and Editors for sending an updated version of the article.

The Authors responded to my comments and introduced corrections according to my suggestions.

I believe that the manuscript in its current version can be published.

Thank you for our collaborative work on this manuscript.